# A Pilot Randomised Controlled Trial to Increase the Sustainment of an Indoor–Outdoor-Free-Play Program in Early Childhood Education and Care Services: A Study Protocol for the Sustaining Play, Sustaining Health (SPSH) Trial

**DOI:** 10.3390/ijerph20065043

**Published:** 2023-03-13

**Authors:** Noor Imad, Nicole Pearson, Alix Hall, Adam Shoesmith, Nicole Nathan, Luke Giles, Alice Grady, Serene Yoong

**Affiliations:** 1School of Health Sciences, Department of Nursing and Allied Health, Swinburne University of Technology, Hawthorn, VIC 3122, Australia; 2Faculty of Health, School of Health and Social Development, Global Centre for Preventive Health and Nutrition, Institute for Health Transformation, Deakin University, Geelong, VIC 3220, Australia; 3Hunter New England Population Health, Wallsend, NSW 2287, Australia; 4School of Medicine and Public Health, College of Health, Medicine and Wellbeing, University of Newcastle, Callaghan, NSW 2308, Australia; 5Hunter Medical Research Institute, New Lambton Heights, NSW 2305, Australia; 6Priority Research Centre for Health Behaviour, University of Newcastle, Callaghan, NSW 2308, Australia

**Keywords:** Early Childhood Education and Care, outdoor free play, physical activity, indoor–outdoor free play, pilot randomised controlled trial, policies, practices

## Abstract

Early Childhood Education and Care (ECEC) settings are important environments to support children’s physical activity (PA). In 2021, COVID-19 regulations recommended the provision of indoor–outdoor free-play programs in ECEC settings to reduce the transmission of COVID-19, resulting in an increased uptake of this practice. As the context has since changed, research suggests that ECEC services could cease the implementation of these practices. Therefore, this pilot randomised controlled trial (RCT) aims to examine the feasibility, acceptability, and impact of a sustainment strategy to ensure the ongoing implementation (sustainment) of ECEC-delivered indoor–outdoor free-play programs. Twenty ECEC services located in New South Wales, Australia that have implemented indoor–outdoor free-play programs since the release of COVID-19 guidelines will be recruited. The services will be randomly allocated either the sustainment strategy or usual care. The “Sustaining Play, Sustaining Health” program consists of eight strategies, developed to address key barriers against and facilitators of sustainment informed by the Integrated Sustainability Framework. The outcomes will be assessed via internal project records, staff surveys, and a self-reported measure of free play. This study will provide important data to support the performance of a fully powered trial within Australian ECEC settings and to inform the development of future sustainment strategies.

## 1. Introduction

Globally, physical inactivity is the fourth leading risk factor for mortality, contributing to approximately 3.2 million deaths each year and accounting for 198.4 disability-adjusted life years (DALYs) per 100,000 people [1,2]. In Australia, 2.5% of the total disease burden is due to physical inactivity and contributes to 10–20% of the individual disease burden from chronic disease such as diabetes, cardiovascular disease, and cancer [1,3]. Additionally, evidence suggests that physical activity (PA) in childhood tends to track into adulthood [4,5,6,7], reducing the risk of chronic diseases later in life [8]. The World Health Organisation (WHO) recommends that children aged 3–4 years should spend at least 180 min active per day, of which at least 60 min should be moderate-to-vigorous physical activity (MVPA) [9]. However, in Australia, only about 6 in 10 (61%) children aged 2–5 met the PA guidelines in 2018 [10]. Similarly, a 2022 meta-analysis of 63 studies from 23 countries found that only 11.26% of pre-schoolers and 10.31% of children adhered to the 24-Hour Movement Guidelines [11]. Therefore, governments internationally recommend the delivery of public health programs and initiatives that focus on increasing the time young children spend being physically active [12].

Early Childhood Education and Care (ECEC) settings are important settings to reach young children with PA-promotion efforts [13,14,15]. As children aged 3–5 years spend an average of 25–35 h a week in ECEC services, these settings provide a broad scope for intervention delivery to increase child activity at the population level [16]. It is recommended internationally and in Australia that ECEC services provide sufficient opportunities for outdoor free play [12]. Further systematic-review and randomised controlled trial (RCT) evidence support the effectiveness of outdoor free-play opportunities in improving child MVPA [17,18]. In particular, indoor–outdoor free-play programs are recommended as ways of increasing outdoor free play [17]. This refers to periods of free play located in both indoor and outdoor environments, with children able to move freely between the two. Our national study with 203 Australian services shows that approximately 60% of ECEC services implemented indoor–outdoor free-play programs between December 2017 and August 2018 [19].

In 2021, in response to the Coronavirus (COVID-19) pandemic, state education authorities in Australia released guidance for ECEC services that included recommendations to services to “consider operating an indoor–outdoor program for the full day/session” and to “encourage appropriate outdoor programs to support social distancing” [20]. This is likely to have resulted in an increased implementation of indoor–outdoor free-play programs. For example, our unpublished data with over 200 Australian ECEC services found that the amount of indoor-only free-play time provided per day reduced from 145 min to 93 min, and that indoor–outdoor free-play programs increased this provision from 71 min to 230 min between 2017/2018 and 2021/2022. Whilst this increase is encouraging, studies examining the effects of public health policies suggest that behaviours are likely to attenuate once acute exposure to the change in stimuli is removed or following the withdrawal of recommendations [21,22,23,24]. Similarly, as the guidance around COVID-19 for ECEC services has now eased [20], we hypothesise that the implementation of indoor–outdoor free-play programs may also attenuate. 

Ensuring indoor–outdoor free-play programs are sustained is crucial to guaranteeing that the ongoing health benefits for children can continue to be achieved [25]. There is, however, a lack of research examining how to sustain the delivery of public health interventions in community settings. Our overview of four Cochrane reviews found no randomised or non-randomised trials that assessed interventions to improve the sustainment of programs targeting PA, nutrition, smoking and alcohol in community-based settings (including ECEC settings, schools, sporting clubs and workplaces) [24]. Therefore, to develop the evidence base and inform the performance of a fully powered trial, we sought to assess the feasibility, acceptability, and impact of a sustainment strategy to improve the continued delivery of indoor–outdoor free-play programs: the ‘Sustaining Play, Sustaining Health (SPSH)’ program.

Specifically, the aims of this research are to assess the:(i)Acceptability, feasibility, and impact of the SPSH program for the sustainment of indoor–outdoor free-play programs (for this trial, defined as the ongoing implementation of evidence-based practice) in ECEC services;(ii)Feasibility of trial methods to inform the performance of a fully powered RCT;(iii)Barriers and facilitators to the sustainment of indoor–outdoor free-play programs in ECEC services.

## 2. Materials and Methods

### 2.1. Design and Setting

The study will employ a 6-month pilot, parallel group RCT design, using guidance on undertaking pilot implementation trials outlined by Pearson et al. [26] and reporting in accordance with the Consolidated Standards of Reporting Trials (CONSORT) pilot guidance [27]. Twenty ECEC services located in New South Wales (NSW), Australia (excluding one region participating in another trial to increase outdoor free play) who report having implemented indoor–outdoor free-play programs for most of the day since July 2021 (i.e., defined as <20% of indoor-only free-play time during core service-operating hours) will be randomised to receive either the SPSH program or usual care. July 2021 was selected as the programs initiation date as this was when guidance around outdoor-free-play programs for ECEC services was released by state education authorities. 

### 2.2. Participants and Recruitment

#### 2.2.1. Sampling Frame

The research team undertook a telephone survey with a randomly selected sample of 231 Long Day Care (LDC) services and preschools in NSW from October 2021 to June 2022, of which 99 services met the eligibility criteria for the current study and form the current sampling frame. These LDC services usually operate from 7 a.m. to 6 p.m. Monday through Friday and cater for children from birth to school age, while preschools cater for children aged 3–5 years and operate from 9 a.m. to 3:30 p.m. [28].

#### 2.2.2. Eligibility

To be eligible to participate, ECEC services needed to report the implementation of indoor–outdoor free-play programs for most of the day (i.e., offering indoor-only free play for only 20% or less of their core operating hours, during which most their children are present in care. This can be defined as approximately 84 min or less, as standard operating hours are between 8:30 a.m. and 3 p.m.) on a typical day since the introduction of the COVID-19 guidance for ECEC services (July 2021) [29]. A typical day represents a day in which excursions or events which may disrupt the service’s usual routine are not planned. Services were considered ineligible if they: (i) had implemented indoor–outdoor free-play programs prior to the introduction of the COVID-19 guidance (July 2021); (ii) catered exclusively for children with special needs; (iii) had a nominated supervisor or staff who did not understand English; or (iv) were a Department of Education community-run service (representing less than 5% of services) across NSW. 

#### 2.2.3. Recruitment Procedures

All services in the sampling frame will receive an invitation email consisting of a recruitment package including an information statement and a link to an online consent form outlining study requirements and requesting participation. Once the consent form has been completed by the nominated supervisor, the online survey will automatically generate the recruitment survey to assess eligibility. A member of the research team will telephone services that have not responded after two weeks of receiving the email to discuss study details, confirm eligibility and request consent for study participation. Recruitment will continue until 20 ECEC services have consented [30,31]. Additionally, the research team will obtain the contact details of an educator who will assist with baseline data-collection processes. 

### 2.3. Randomisation and Blinding

Following baseline data collection, services will be randomly allocated to either receive the SPSH program or usual care. A statistician will set up block randomisation using a computerised random number generator. Block randomisation will be used to ensure group allocation is approximately equal. This will be uploaded onto the online data-capturing platform (REDCap) [32,33], where service records are stored and concealed from statisticians. This pilot RCT will be conducted as an open trial as the sustainment strategy cannot be delivered without revealing the allocation to services. Therefore, services and data collectors will not be blinded to group allocation; however, those conducting data analysis will remain blinded to allocation for the main impact outcome (minutes of outdoor free play).

### 2.4. Intervention

#### 2.4.1. Theoretical Framework

The development of the 6-month sustainment strategy to increase sustainment of indoor–outdoor free-play programs was overseen by an advisory group consisting of implementation and behavioural scientists, health-promotion staff and health policymakers; it was informed by the Integrated Sustainability Framework developed by Shelton et al. [25]. The Integrated Sustainability Framework is a comprehensive, theoretically informed sustainability-determinants framework developed based on available empirical research on factors identified as influencing intervention sustainment across a range of contexts. The framework outlines the dynamic interactions between 21 multi-level factors across five domains, including: Outer Contextual Factors, Inner Contextual Factors, Processes, Characteristics of the Interventionists and Population and Characteristics of the Intervention [25]. To identify the determinants related to the sustainment of PA programs, data were collected from a telephone survey with ECEC services nationally assessing the Integrated Sustainability Framework constructs and extracted from a systematic review of sustainability determinants within ECEC and school settings [34].

From this data, we identified 20 barriers associated to the sustainment of PA programs in the ECEC and school settings, which mapped to all five domains of the Integrated Sustainability Framework. This resulted in the development of eight strategies (see Table 1) that sought to address these barriers and support the sustainment of implementing indoor–outdoor free-play programs. As barriers were predominantly mapped to the outer contextual factors (e.g., lack of future external funding/financial support, or lack of state requirements) and processes domains (e.g., lack of training/professional development opportunities to upskill, or lack of collaboration with community groups), these domains were of particular focus when developing the strategies. The Expert Recommendations for Implementing Change (ERIC) glossary adapted to describe strategies to address sustainment [35] was used to select the strategies, including: (i) identifying an opinion leader to act as a key driver of the sustainment strategy; (ii) affirming the service’s intent to continue; (iii) providing the service with technical support and educational resources to self-direct the sustainment strategy; (iv) developing a formal sustainment blueprint; (v) engaging with family and community members; (vi) reviewing and changing service policy to integrate indoor–outdoor free-play programs; and (vii) developing an ongoing monitoring plan to self-direct the sustainment of the evidence-based practice (EBP) well after the intervention period [36]. The intervention-mapping process is outlined in Table 1, below.

#### 2.4.2. Sample Size

As this is a pilot study, a formal sample-size calculation on efficacy was not performed, in line with best-practice guidance [27]. However, the researchers sought to recruit 20 ECEC services, as this number was considered feasible in the timeframe and sufficient to provide an adequate indication of the feasibility and acceptability of study methods and the sustainment strategy [30,31].

#### 2.4.3. Intervention Components

Following baseline-data collection and randomisation, services allocated to the intervention group will receive the sustainment strategy. Specifically, services will be supported to continue to implement indoor–outdoor free-play programs.

The delivery of the SPSH program will include a minimum of four remotely delivered scheduled contacts, including two meetings via video calls and two support emails (see Table 2). The use of video calls has been found to be an effective and acceptable modality of delivering behaviour-change interventions for this type of setting as an alternative to face-to-face contact [56,57]. Contacts will target the nominated supervisor and an educator responsible for supporting the implementation of PA programs in the ECEC service. The intervention will be delivered by experienced health-promotion officers (HPO) employed by a local health-promotion unit with tertiary qualifications in physical activity, health promotion, nutrition and dietetics. They are employed by the local health services and have extensive experience supporting ECEC services in the delivery of healthy-eating and physical-activity programs. Additionally, the HPOs have undertaken behaviour-change training with an expert behavioural scientist to support behaviour change in this setting.

(a)Identify opinion leaders at the service:

ERIC Strategy: *Engage with local opinion leaders (i.e., supervisors or room leaders)*

Barriers Targeted: *Lack of administrative buy-in and support/leadership/management and lack of program leaders/facilitators/champions.*

Engaging with opinion leaders is an important strategy to understand the existing internal support for the EBP within the service and is a key driver of implementation in ECEC settings [58,59,60,61]. The ECEC services will be asked to identify any colleagues who they believe are drivers of implementation or “educationally influential”. These may be the nominated supervisor or another existing staff member, such as a room leader. If this is not the nominated supervisor, this person will be invited to participate in service contacts along with the nominated supervisor/service manager. The HPO will determine the existing roles and influence of these staff members in current implementation processes as a component of the first 30–60-min introductory-meeting video-call contact. The SPSH program will be discussed and additional roles or tasks for these key staff members will be co-developed as part of the process of undertaking a “Sustainability Action Plan” (see strategy d for further detail). 

(b)Affirm intent of continuity:

ERIC Strategy: *Affirm formal commitments*

Barriers targeted: *Lack of motivation/interest, limited cost-effectiveness/feasibility of program, time required to implement and record uptake of the program and competing resources, responsibilities and curriculum demands.*

This strategy aims to assess whether commitments are being upheld and whether new commitments are required to help sustain the intervention [35]. Services will be supported in affirming the service’s intent to continue and readiness for the long-term implementation of the EBP. To this end, the HPO will explicitly ask the nominated supervisor and the opinion leader questions regarding their intention to continue the implementation of their current indoor–outdoor free-play programs during the introductory-meeting video call and 3-month support-meeting video call. To explore the likelihood of the services being able to sustain long-term change, the HPO will also ascertain the motivations behind the services’ original implementation, determine any barriers to and facilitators of the ongoing implementation of EBP and the readiness to implement the SPSH program to enhance integration into typical care. As part of supporting a commitment to long-term change, the HPO will help services draft ideas for a formal “motivational statement” for the implementation of the EBP. The services will be helped to use this as part of their staff and community communications, as well as for inclusion in policy and procedure updates, as appropriate.

(c)Provide local technical assistance to support integrating strategies:

ERIC Strategy: *Provide local technical assistance*

Barriers targeted: *Lack of training/professional development opportunities to upskill and lack of clear data on effectiveness of program.*

This strategy was previously found to support the implementation of ECEC-based interventions and to be effective in improving implementation, staff motivation and problem solving within these interventions [62,63,64]. The HPOs will provide local technical assistance in the form of two 30–60-min video calls from the HPO and a minimum of two action-oriented emails throughout the intervention period to assist them in implementing the SPSH program. Video calls will occur at months one and three, with policy reviews via email sent out at months two and four–five. Each contact will draw on continuous quality-improvement principles to review progress on the “Sustainability Action Plan” and provide feedback on the integration of sustainment strategies. The first video call will occur after recruitment and will focus on introducing the SPSH program, explaining the roles of the opinion leader and the nominated supervisor (see a), understanding any determinants of sustainability for the service (see b) and discussing the development of the “Sustainability Action Plan” (see d). The second video call will take place at the 3-month time point and will focus on the service’s progress with its “Sustainability Action Plan” and any challenges experienced. The two emails will consist of providing feedback on the service’s policy through the use of a template and suggestions for amendments (see g). Additional support contacts may occur at time points outside of those specified if further support is required and will be recorded as part of adaptations to the SPSH program. 

(d)Develop a formal sustainment blueprint:

ERIC Strategy: *Develop a formal implementation blueprint*

Barriers targeted: *Lack of centralised coordination and absence of plans with defined measures.*

Similar strategies haves been used in previous implementation interventions to facilitate improvement in implementation within ECEC services [65]. Services will be supported to develop an action plan (blueprint) for the service to implement the SPSH program. This will support the identification of strategies to be put in place as well as sustainment strategies that may have already been undertaken (fully or partially). Examples of strategies include notifying families, community members and staff of indoor-outdoor free play programs, updating the service policy, and conducting indoor-outdoor free play programs as part of the daily routine. Progress with the plan will be monitored during the 3-month support meeting video call, and adjustments and/or support will be provided to help meet strategies, as required.

(e)Distribute educational materials to support ongoing implementation:

ERIC Strategy: *Distribute educational materials*

Barriers targeted: *Lack of future external/financial support, lack of state requirements, lack of evidence-based treatments, government perceptions, lack of training/professional development opportunities to upskill and lack of clear data on effectiveness of program.*

Distributing educational materials to services has been found to be highly acceptable among ECEC service staff and is commonly used as part of multicomponent implementation interventions [65,66,67]. Services will be provided with a resource pack approximately 3 days prior to the introductory meeting video call. It will consist of resources developed or adapted from materials previously used by the research team to increase outdoor free-play opportunities. The resources are intended to target barriers to sustainment and support implementation of the “Sustainability Action Plan”. Resources include fact sheets on external organisations that support outdoor free play, how outdoor free play aligns with the sector-accreditation standards, a summary of the COVID-19 guidelines relating to outdoor play, Australian PA guidelines, family and staff templates to include as part of orientation packages, a service-policy template and tips for maintaining indoor–outdoor free-play programs across all seasons. Further resources will be developed and provided, tailored to the service’s specific needs. 

(f)Engaging with family members:

ERIC Strategy: *Involve patients/consumers and family members*

Barriers targeted: *Lack of collaboration with community groups, communicating information to stakeholders, lack of parental buy-in/support and staff turnover.*

Engaging with family members and communities is critical to support initial adoption and implementation of programs [45,68]. This strategy will include the provision of resources such as 6 newsletter-communication snippets providing information around the benefits of outdoor free play, outdoor play in various seasons and outdoor play and COVID-19. These are to be used in service-provided family newsletters or other communication channels at the service’s discretion, with a minimum requirement of 1 newsletter snippet to be sent out throughout the 6-month SPSH program. A template for the inclusion of indoor-outdoor free play programs in the family-orientation package will also be provided. Services will be asked to distribute these using their usual communication channels.

(g)Reviewing and embedding change into policy:

ERIC Strategy: *Mandate change*

Barriers targeted: *Absence of plan with defined measures, lack of centralised coordination and lack of training/professional development opportunities to upskill.*

Implementing a written policy around PA has been associated with higher levels of PA [69,70,71]. This strategy aims to ensure that the commitment to sustain indoor–outdoor free-play programs is reflected in a relevant service policy. At the 2-month- and 4–5-month-timepoint contacts, the HPO will review each service’s policy and suggest potential changes to promote the continuation of indoor–outdoor free-play programs.

(h)Develop an ongoing monitoring strategy:

ERIC Strategy: *Develop and implement tools for quality monitoring*

Barriers targeted: *Accuracy in assessing impact of program, lack of clear data on effectiveness of program and absence of plan with defined measures.*

Ongoing monitoring is important to allow services to continue to determine whether they are implementing the program as intended [72]. Services will be encouraged to develop their own plan during the intervention contacts with the HPOs to ensure the EBP continues to be implemented long term. Alternatively, services can continue to use a tool provided by the research team as a way of continuing to document implementation twice a year. 

#### 2.4.4. Control Group and Contamination

The delivery of intervention components will be under the control of the research team and will not be provided to the control-group services during the intervention period. The ECEC services in the control group will receive ‘usual care’. This includes the provision of a generic email with a link to the benefits of outdoor free play provided by a HPO [29]. 

### 2.5. Data Collection and Measures

Baseline data were collected in September–December 2022, while follow-up data collection will occur at approximately 6 months (primary timepoint, as it will occur approximately 2 years after COVID-19 guidelines) and 12 months post-baseline. The following data will be collected using either a telephone or online survey undertaken by a member of the research team. 

The nominated supervisor/service manager (or the opinion leader responsible for implementation of the SPSH program) will be asked to report on the feasibility and acceptability of the SPSH program, as well as the factors that influence the ongoing delivery of indoor–outdoor free-play programs in their service setting (Table 3). In addition, the Free Play Record (FPR) (a measure developed by the research team to assess service implementation of indoor–outdoor free-play programs and number of minutes of outdoor play) will primarily be completed by the most appropriate person or a person with knowledge of the performance of free play in the service (such as the opinion leader or nominated supervisor/service manager). 

#### 2.5.1. Outcomes

*(i)* *Acceptability, feasibility, and potential impact of the SPSH program on the sustainment of indoor–outdoor free-play program (defined as ongoing implementation of the EBP) in ECEC services*;


*(1) Acceptability of the SPSH program:*


This is defined as ECEC service-staff perceptions of the extent to which the SPSH program is agreeable, palatable, or satisfactory [73].

During the 6-month post-baseline follow-up telephone or online survey, the service manager at the intervention services will be asked to report on 8 items capturing information related to the acceptability (e.g., attitude towards the SPSH program, burden, ethicality of the program, coherence, effectiveness, self-efficacy) of the SPSH program. These items are underpinned by the Theoretical Framework of Acceptability (TFA), a validated framework used to assess the acceptability of healthcare interventions from the perspectives of intervention deliverers and recipients [74].

This survey consists of 7 items measured on a Likert scale with response options ranging from 1 to 5. The domains include their affective attitude towards SPSH (“Did you like or dislike SPSH?”), the effort involved, or “Burden” (“How much effort to engage in SPSH?”), the perceived effectiveness of SPSH (“SPSH has helped to continue the sustainment of indoor–outdoor free-play programs?”), intervention coherence (“is it clear how SPSH has helped to continue the sustainment of indoor-outdoor free play programs”), self-efficacy (“How confident did you feel about participating in SPSH?”), opportunity costs (“Participating in SPSH interfered with other priorities”) and general acceptability (“How acceptable was SPSH?”). Generally, 1 is a negative response/disagreement with the statement, while 5 is a positive response/agreement with the statement, except for Burden (domain 2) and Opportunity costs (domain 6), where 1 is a positive response, while 5 is a negative response. The 8th item is an open-ended question to capture any further comments on the acceptability of the SPSH program. 


*(2) Feasibility of the SPSH program:*


This is defined as ECEC services’ perceptions of the extent to which the SPSH program can be successfully used or carried out at scale within different organisations or settings [73].

At the 6-month post-baseline follow-up time point, the intervention services will be assessed via a survey using 4 items (3 questions on a Likert scale and 1 open-ended item) adapted from the Feasibility of Intervention measure [75]. This questionnaire will capture information related to the feasibility (continuing to implement the SPSH program seems possible, easy and doable) of the SPSH program and its fit within the service. The nominated supervisor/service manager will be asked to rate their level of agreement with each item on a scale of 1 (Strongly disagree) to 5 (Strongly agree), along with an opportunity to provide any further comments on the feasibility of the SPSH program via the open-ended item. 


*(3) Sustainment of indoor–outdoor free-play programs (potential impact):*


Sustainment as an outcome is defined as: “The extent the innovation is in place or being delivered over the long-term” [76]. The delivery of indoor–outdoor free-play programs will be assessed at baseline and during the follow-up, at 6 months and 12 months post-baseline. In this study, it will be interpreted and applied as the number of minutes of indoor-only free play provided by the services over 5 consecutive days.

Data on the number of minutes that children are provided with indoor-only, indoor–outdoor, and outdoor-only free play from a randomly selected week will be collected via the FPR. The FPR is adapted from existing ECEC measures of outdoor play: Environment and Policy Assessment and Observation—Self-Report (EPAO—SR) and Nutritional and Physical Activity Self-Assessment for Child Care (NAPSACC) [77,78]. The adapted FPR measure has been pilot-tested with ECEC services and has been reported to take less than 5 min/day on average and to be acceptable and easy to complete. Further, our unpublished data indicate that the Free Play Record overestimates opportunities for outdoor free play (the primary outcome) by just 7.6 min (95% CI −15, 15) compared to direct observations (*p* < 0.3651). The FPR will be provided to the nominated supervisor/service manager or educators at each ECEC service electronically, via email, by the research team, at baseline and during the follow-up period. The educators/nominated supervisor(s) will be asked to complete the FPR for each session of free play throughout the day across the selected week (5 consecutive days) and return the completed FPR to the research team. If not returned, a follow-up email or phone call will then be made to the appropriate delegate to follow up on the progress of FPR completion. When the research team receive the completed FPR, a phone call will be made to the ECEC service to obtain any missing data, clarify non-legible responses, if required and assess any specific challenges to completing the FPR. The research team have successfully undertaken this with 90 services as part of previous trials [79].


*(4) Adoption of SPSH-program recommendations:*


Adoption is defined as the intention, initial decision, or action undertaken to employ an innovation or evidence-based practice [80]. This will be assessed according to how many strategies within the “Sustainability Action Plan” are completed by the 6-month time point, to determine how many sustainment strategies were applied according to the intentions of the SPSH program. 

*(ii)* 
*Feasibility of use of trial methods to inform the conduct of a fully powered RCT*



*(1) Feasibility of the trial methods:*


In order to inform a fully powered RCT and to assess whether the trial methods are feasible and replicable, project records will be used to report consent rate, percentage of missing data from completion of the FPR (internal self-reported measure), comparison of consenter/non-consenter characteristics, attrition rate, withdrawal and reasons and contamination [26,27]. 


*(2) Fidelity to delivering the SPSH program:*


Fidelity is defined as the degree to which the SPSH program was implemented as described or intended in the original protocol [80]. Fidelity will be assessed using project records, which will determine the extent to which each sustainment strategy was delivered by the HPO. This will be undertaken by the research team documenting these data during recruitment and intervention delivery on a spreadsheet. 

*(iii)* 
*Barriers to and facilitators of the sustainment of indoor–outdoor free-play programs in ECEC services*



*(1) Barriers to and facilitators of indoor–outdoor free play:*


During the baseline and 6-month follow-up telephone/online surveys with the nominated supervisor/service manager, barriers, and facilitators to sustaining the EBP will be assessed. This will be conducted using an adapted measure of the 29-item scale, which measures sustainability determinants across the constructs of the Integrated Sustainability Framework (outer and inner contextual factors, processes, and characteristics of the intervention) developed by the team. This adapted 9-item scale measures the outer contextual factors and processes that are perceived to be influential in intervention sustainment in ECEC settings. The nominated supervisor/service manager will be asked to rate their level of agreement with a number of items on a scale from 1 (completely disagree) to 5 (completely agree). 


*(2) ECEC service characteristics*


Data regarding ECEC service characteristics, including service type, age group of children at the service, operating hours, number of children enrolled aged three to six years and number of educators at the service, will be collected via a baseline survey of ECEC nominated supervisors/service managers and educators. Service locations’ postcodes will be used to determine socioeconomic status (SES) and rurality. Services with postcodes ranked in the top 50% of state postcodes based on the 2016 Socio-Economic Indexes for Areas (SEIFA) [81] will be classed as “higher socioeconomic areas”, whereas those in the lower 50% will be classed as “lower socioeconomic areas”. 


*(3) Cost*


Similar to a previous analysis [82], the direct cost of each sustainment strategy delivered by the HPOs, including labor (i.e., HPO preparation, administration and delivery of the program) will be calculated. Service-delivery costs will be recorded by the HPOs delivering the interventions. Costs in AUD, 2022/2023 will be calculated by multiplying the time spent (in hours) on each sustainment strategy by the hourly wage rate of HPOs delivering the intervention. The cost for nominated supervisor/educators to receive the SPSH program delivered by the HPOs and time taken to implement the strategies will also be calculated. Similar to previous studies examining the cost of receiving interventions within ECEC settings [83], costs will be calculated and multiplied by the time spent (in hours) receiving the SPSH program by the estimated hourly wage rate of the nominated supervisors/service managers and Educators [84]. The time spent will account for the services representatives’ attendance at all meetings and implementation of the strategies in their respective service settings. 

#### 2.5.2. Overall Data Management 

Management of trial data will be in accordance with a data-management protocol, developed and approved by the nominated advisory group. Data will be stored in accordance with the requirements of the Hunter New England (HNE), University of Newcastle, and Swinburne University of Technology Human Research Ethics Committees (HRECs). All questionnaires and notes will be deidentified and recordings will be stored in a locked HNE Microsoft Teams folder to maintain confidentiality. 

### 2.6. Statistical Analyses

All statistical analyses will be undertaken using SAS v9.2 statistical software. The main data-collection point will be at 6 months post-baseline, as this will be approximately 2 years from the initial introduction of the COVID-19 guidelines. The 12-month timepoint will be used for assessing whether indoor–outdoor free-play programs continue to be delivered following cessation of active sustainment strategy (the SPSH program). For all outcomes, we will assess for normality and use non-parametric test where non-normality is observed.

#### 2.6.1. Acceptability and Feasibility 

Descriptive statistics (mean and standard deviation (SD) or median and range when non-normal) will be generated for both acceptability and feasibility at 6 months post-baseline. For acceptability, individual scores from each domain will be summed (reverse scored where necessary) and used to provide a score for each service. For example, ≤2 will be a negative response, indicating the SPSH program is considered unacceptable, while ≥4 will be a positive response, indicating the SPSH program is considered acceptable. In cases of missing values, we will impute the average scores. Answers to the open-ended question will be analysed qualitatively, using thematic analysis to identify key themes. For feasibility, the individual scores of each of the three items (“the strategy seems possible”, “seems doable” and “seems easy”) will be summed to provide an overall score for each service, in which ≥4 (higher scores) will indicate greater feasibility, while ≤2 (lower scores) will indicate lower feasibility, consistent the recommendations of Weiner et al. (2017) in the Feasibility of Intervention Measure. For the open-ended question, the responses will be analysed qualitatively, using thematic analysis to identify key themes. 

#### 2.6.2. Sustainment of Indoor–Outdoor Free Play

Separate analyses will be performed at each follow-up time point (6 and 12 months post-baseline) for each primary outcome, total minutes of indoor-only free play and total minutes of indoor-outdoor free play. Linear regression models will be used to compare intervention and control group differences, controlling for the baseline outcome data. Alternatively, equivalent non-parametric tests will be performed if linear regression assumptions are not met.. Analyses will follow the intention-to-treat principle and include all data analysed according to the group to which each service was originally randomly assigned. Imputation techniques will be applied where appropriate.

#### 2.6.3. Adoption of SPSH-Program Recommendations

Adoption will be assessed by counting how many strategies were “complete” out of the minimum of 4 strategies in the Sustainability Action Plan during the 6-month program for each service. The minimum requirement for services is to complete at least one goal from the “Notify families and community members of indoor–outdoor free play” strategy, “Notify staff of indoor–outdoor free play” strategy, “Update policy” strategy and “Conduct indoor–outdoor free play as part of daily routine” strategy. The number of completed strategies will be summed to determine how many strategies were completed by each service. A minimum of 4 strategies set as “complete” will be required for the service to have adopted the SPSH program recommendations. After all intervention services have had their action plan counted, a mean will be calculated, showing the average number of services adopting the SPSH program recommendations. 

#### 2.6.4. Feasibility of Trial Methods

Feasibility of the trial methods will be analysed at baseline and 6 months post-baseline. Consent rate, percentage of missing data from completion of the FPR, comparison of consenter/non-consenter characteristics, attrition rate, withdrawal and reasons and contamination will all be analysed using descriptive statistics. 

#### 2.6.5. Fidelity to Delivery of the SPSH Program

Fidelity will be assessed using an internal spreadsheet tracking recruitment and intervention delivery. Each step records time spent, staff involved and whether it has been completed. Analysis will include summing the number of completed steps according to the protocol for each service to determine the degree of implementation according to the original protocol. 

#### 2.6.6. Barriers and Facilitators to the Sustainment 

Barriers and facilitators of sustainment will be analysed at 6 months post-baseline. The outer contextual factors and processes will be scored individually. The total scores (i.e., the mean and SD) of each of the two domains will be calculated. The total scores will be calculated by summing all scores for all items within the subdomain and dividing by the total number of responses within the domain. The higher domain scores (≥4) will be considered facilitators of sustainment, while the lower domain scores (<4) will be considered barriers to sustainment.

#### 2.6.7. Progression Criteria

An advisory group will assess suitability for progression to a fully powered trial using data collected around feasibility of trial methods and outcomes related to potential sustainment, fidelity, and acceptability. These decisions will be made via majority, by core members of the research team, including a representative from a health service that intends to adopt the sustainment strategy if it is found to be beneficial. Specifically, the team must deem the intervention and sustainment strategy to be sufficiently acceptable and feasible for it to be likely to be adopted by approximately 25% of ECEC services to which it is offered (i.e., a consent rate of >25%) [85]. Alternatively, the team may decide that this could be reasonably expected with adaptations to the intervention or sustainment approach, based on steps previously employed [86]. Further, as recommended, measures of delivery of the SPSH program, together with acceptability and feasibility, as well as barriers and facilitators, will be used to identify opportunities to further strengthen the intervention and refine trial methods prior to a fully powered implementation trial.

## 3. Trial Status

The trial is currently in the intervention-delivery stage. It is not anticipated that any events will occur that would warrant discontinuing the trial as the SPSH program will be delivered completely online. Any unforeseen adverse events would be reported to the HNE HREC (primary approval committee), and appropriate action would be taken to address the event. The trial’s registration record will be updated with any protocol modifications and any deviations from the original protocol will be reported when publishing trial outcomes.

## 4. Discussion

This protocol outlines the development and evaluation of a theory-informed sustainment strategy to increase the sustainment (ongoing implementation) of an evidence-based PA practice (indoor–outdoor programs in ECEC settings). While the sustainment of PA practices in ECEC is needed for continued health improvements in the population and are likely to provide considerable cost savings, there is an absence of empirical evidence on how this can be achieved. This pilot trial seeks to describe, for the first time, the feasibility, acceptability and impact of a program (SPSH) on the sustainment of EBP, providing an insight into how to support the sustainment of health-promotion programs delivered in the ECEC context.

## 5. Conclusions

This protocol outlines the methods and development of a strategy to improve the sustainment of outdoor-free-play opportunities through ECEC services. The application of the strategy will provide important proof-of-concept data to support the conduct of a fully powered trial, the first of its kind, as well as informing the development of future sustainment strategies in ECEC settings more broadly.

## Figures and Tables

**Table 1 ijerph-20-05043-t001:** Mapping of sustainment strategies against the Integrated Sustainability Framework and the Action, Actor, Context, Target, Time (AACTT) Framework [25,37].

Sustainment Strategy	Strategy Description (*ERIC Strategy and Definitions*) [35]	Actor(s)/Personnel	Context	Target	Time	Integrated Sustainability Framework Domain + Factor [25]	Barriers Addressed [34]	Justification
Identify opinion leaders at the service	*ERIC Strategy: Engage with local opinion leaders.* *Periodically engage with providers identified by colleagues as opinion leaders or “educationally influential” about the importance of continuing to deliver the practice innovation in the hopes that they will influence colleagues to sustain its use.* The identification of and engagement with a staff member who could be considered as a key driver or influencer in the original implementation of the EBP.	Health-promotion officer/nominated supervisor	Phone call	Opinion leader	Prior to the introductory-meeting video call	Inner Contextual Factors *Organisational leadership and support*	Lack of administrative buy-in and support/ leadership/management ^a^ [38,39,40,41,42,43,44]	The identification of an opinion leader is intended to target these barriers around leadership/support from management by identifying a staff member at the service who can lead the sustainment strategy to encourage and motivate other staff to continue to deliver indoor-outdoor free play programs.
						Inner Contextual Factors *Programme champions*	Lack of programme leaders/facilitators/champions ^a^ [38,43]	
Affirm intent of continuity	*ERIC Strategy: Affirm formal commitments.* *Revisit the written commitments obtained from key partners that state what they will do to implement and sustain the innovation. Assess whether these commitments are being upheld and whether new commitments are required to help sustain the innovation.* The level of commitment to ongoing implementation will also be ascertained by establishing service motivations for initial and continued implementation, as well as understanding any barriers and facilitators to the long-term implementation of the EBP and the sustainment strategies. This will be developed using the following strategies: Sustainability Action Plan; Motivational statement.	Health-promotion officer	Video calls/phone calls	Opinion leader	Introductory-meeting video call Three-month support meeting Additional contacts upon request	Characteristics of interventionists and population *Implementer characteristics*	Lack of motivation/interest ^a^ [41,44,45,46,47]	Affirming the services intent to continue helps to tackle the lack-of-motivation barrier by regularly reminding the service of the importance of indoor–outdoor free-play programs and their benefits to children.
						Characteristics of the intervention *Perceived benefits*	Limited cost-effectiveness/feasibility of program ^a^ [41] Time required to implement and record uptake of the program ^a^ [45] Competing resources, responsibilities and curriculum demands ^a^ [39,41,43,44,45,48]	This strategy also targets the barriers to feasibility of the EBP, the time required to implement the EBP and competing responsibilities, as it focuses on re-iterating the services’ intent to continue their practice of delivering indoor–outdoor free play programs. This is important as it highlights that the service is already implementing the EBP; thus, this strategy of the SPSH program aims to aid the services to continue their current practices rather than change routines or add to staff workload.
Provide local technical assistance to support integrating strategies	*ERIC Strategy: Provide local technical assistance.* *Develop and use a system to deliver technical assistance focused on implementation and sustainment issues using local personnel.* Opinion leader will receive 2 video-call-support meetings and a minimum of 2 emails throughout the intervention period. The initial video meeting will highlight the purpose and sustainment strategies. Each contact will review progress and provide feedback on integration of sustainment strategies. Additional support will be provided as required.	Health promotion officer	Introductory/3-month support meeting: Video call Other intervention contacts (as required): phone/email	NS/Opinion leader	One video call at beginning of intervention and at the 3-month timepoint One email at the 2- and 4/5-month timepoints	Processes ^b^ *Training/supervision/support*	Lack of training/professional development opportunities to upskill ^a^ [41,42,43,46,47,49,50]	The provision of technical support aims to target this barrier through providing services with the skills/tools thar are needed to continue to implement the EBP themselves without external support.
						Processes ^b^ *Programme evaluation/data*	Lack of clear data on effectiveness of program ^a^ [45]	In the case of clarity around the effectiveness of the EBP, technical support is provided to develop a motivational statement supporting the delivery of the EBP, to ensure services understand the importance of this practice. In addition, this strategy aims to address any challenges experienced when delivering the EBP.
Develop a formal sustainment blueprint	*ERIC Strategy: Develop a formal implementation blueprint.* *Develop a formal blueprint which includes all goals and strategies to guide the implementation effort over time.* This is the introduction of an action plan (“Sustainability Action Plan”) for the service to implement best-practice sustainment strategies. This will track the implementation of the sustainment strategies and their progress.	Health promotion officer and Opinion leader	Introductory/3-month support meeting: Video call	Opinion leader	Provided: prior to introductory support call Discussed: during the introductory and 3-month support meeting Action: self-directed throughout the 6-month intervention period	Inner Contextual Factors *Organisational leadership/support*	Lack of centralised coordination ^a^ [39]	This aims to address these barriers by developing a Sustainability Action Plan, with defined strategies, to support the ongoing implementation of the EBP in the long term. The action plan is intended to be a self-directed tool that services use to track their progress in implementing the sustainment strategies, for which the opinion leader is responsible. This strategy allows more centralised leadership, led by the service, rather than externally.
						Processes ^b^ *Communications and strategic planning*	Absence of plan with defined measures ^a^ [45]	
Distribute educational materials	*ERIC Strategy: Distribute educational materials.* *Distribute educational materials (including guidelines, manuals, and toolkits) in person, by mail, and/or electronically.* A resource pack will be provided to services, which will include the following resources: Which external organisations support outdoor free play? How outdoor free play aligns with the National Quality Framework; COVID-19 outdoor-play guidance; Australian Physical Activity guidelines; Indoor–Outdoor Free Play in any Weather fact sheet; Benefits of Outdoor Free Play fact sheet; Shade and Heat fact sheet.	Health promotion officer	Email	Opinion leader	Provided: prior to introductory support meeting Discussed: during introductory support meeting	Outer Contextual Factors ^b^ *Funding environment and availability*	Lack of future external funding/financial support ^a^ [39,40,44,45,51,52]	The resource pack includes a fact sheet on a number of organisations that help ECEC services to deliver and encourage outdoor free play.
						Outer Contextual Factors ^b^ *Sociopolitical context*	Lack of state requirements ^a^ [47,53,54]	The resource pack includes the requirements of outdoor free play in the National Quality Framework and the Early Years Learning Framework, which are part of the accreditation standards for ECEC services in Australia.
						Outer Contextual Factors ^b^ *Values, needs, priorities*	Lack of evidence-based treatments ^a^ [39] Government perceptions ^a^ [45]	The “benefits of outdoor free play” fact sheet highlights the importance of the EBP through an evidence-based lens. Government perceptions are addressed through the fact sheets on the Australian government supporting outdoor free play through the Australian physical activity guidelines and the NSW COVID-19 outdoor-free-play guidance for ECEC services.
						Processes ^b^ *Training/supervision/support*	Lack of training/professional development opportunities to upskill ^a^ [41,42,43,46,47,49,50]	The Indoor–Outdoor Free Play in any Weather fact sheet and the Shade and Heat fact sheet highlight the different ways in which children can enjoy the outdoors in different seasons while staying safe. This strategy addresses this barrier by exploring the different ways in which indoor–outdoor free-play programs can be implemented at an ECEC service.
						Processes ^b^ *Programme evaluation/data*	Lack of clear data on effectiveness of program ^a^ [45]	These barriers are addressed through the Benefits of Outdoor Free Play fact sheet, highlighting the importance of outdoor free play for children due to the many benefits children receive. This EBP is strongly considered in the literature to be an effective way to increase PA, which in turn has many health benefits for children.
Engaging with families	*ERIC Strategy: Involve patients/consumers and family members.* *Engage or include patients/consumers and families in the implementation and sustainment efforts.* The following resources will be provided to services: Orientation-package templates for new families and staff; Template for advocating the EBP on the services’ website; Newsletter snippets for families.	Health-promotion officer/ Opinion leader	Email/Video call	Families /staff at the service Community members	Provided: prior to introductory support meeting Discussed: during introductory support meeting Action: self-directed throughout the 6-month intervention period	Processes ^b^ *Partnership/* *engagement*	Lack of collaboration with community groups ^a^ [43]	The use of the newsletter snippets and orientation packages for families allow the services to communicate with families and the community regarding the delivery of indoor–outdoor free-play programs in their respective service setting. This communication opens up opportunities for families to weigh in and provide feedback, as well as to promote the benefits of this practice to community members. In this way, barriers such as lack of collaboration with communities, communicating information and lack of parental buy-in are addressed.
						Processes ^b^ *Communications and strategic planning*	Communicating information to stakeholders^a^ [51]	
						Characteristics of the interventionists and population *Population characteristics*	Lack of parental buy-in/support (e.g., uninterested parents) ^a^ [41,43,44,45]	
						Inner Contextual Factors *Organisational stability*	Staff turnover ^a,c^ [22,38,41,44,55]	By advocating for the delivery of indoor–outdoor free-play programs on the services’ websites and the provision of orientation packages for new staff, the barrier of staff turnover is addressed, as new staff members are aware of the current practices and, therefore, should continue to uphold these as new members of their respective teams.
Review and embedding change into policy	*ERIC Strategy: Mandate change.* *Have leadership declare the priority of the innovation and their determination to have it implemented and sustained.* The following resources will be provided to services: Policy template for indoor-outdoor free play programs; Policy checklist.	Health-promotion officer/ Opinion leader	Email	Staff at the service/Service routine/Service policy	Provided: prior to introductory support meeting Discussed: during the 2-month and 4-month policy review email Action: self-directed throughout the 6-month intervention period	Inner Contextual Factors *Organisational leadership/support*	Lack of centralised coordination ^a^ [39]	To address these barriers, the provision of a policy template that outlines the benefits of indoor–outdoor free-play programs and their consistent provision enables services to implement changes to their service policies. By allowing services to formally integrate the EBP into their service policies, staff members will be able to develop a set of procedures to ensure the provision of indoor–outdoor free-play programs as part of their daily routine.
						Processes ^b^ *Training/supervision/support*	Lack of training/professional development opportunities to upskill ^a^ [41,42,43,46,47,49,50]	
	Reviewing and embedding the practice of indoor–outdoor free-play programs in the service routine as a strategy on the “Sustainability Action Plan”.	Health-promotion officer/Opinion leader	Email	Staff at the service/Service routine/Service policy	Provided: prior to introductory support meeting Discussed: during the 2-month and 4-month policy review email Action: self-directed throughout the 6-month intervention period	Processes ^b^*Communications and strategic planning*	Absence of plan with defined measures ^a^ [45]	The “review policy” strategy in the Sustainability Action Plan will allow services the opportunity to review their current service policies and embed the regular practice of providing indoor–outdoor free-play programs as part of their daily routine. This strategy will provide services with a plan to ensure the provision of indoor–outdoor free-play programs in the long term.
Develop and implement tools for quality monitoring	*ERIC Strategy: Develop and implement tools for quality monitoring. Develop, test, and introduce into quality-monitoring systems the right input—the appropriate language, protocols, algorithms, standards, and measures (of processes, patient/consumer outcomes, and implementation outcomes) that are often specific to the innovation being implemented and sustained.* At the initial contact, an “ongoing monitoring plan” item will be added to the Sustainability Action Plan.	Health-promotion officer/Opinion leader	Email/Video call	Staff at the service	Provided: prior to introductory support meeting Discussed: during introductory support meeting and 3-month support meeting Action: self-directed throughout the 6-month intervention period	Processes ^b^ *Programme evaluation/data*	Accuracy in assessing impact of program ^a^ [45,51] Lack of clear data on effectiveness of program ^a^ [45]	The use of an ongoing monitoring tool, such as the FPR, will enable services to monitor their ongoing delivery of the EBP. This tool will highlight the gaps in indoor–outdoor free-play provision and address the lack of professional-development opportunities to upskill. In this way, staff will have the opportunity to self-monitor and adjust their practices. By using an ongoing monitoring tool, services will be able to accurately assess the impact of the program through child benefits, family feedback or other measures, thus enabling the identification of the effectiveness of indoor–outdoor free-play programs.
						**Processes** ^b^ *Communications and strategic planning*	Absence of plan with defined measures ^a^ [45]	

^a^ Barriers identified from interventions in schools [34]; ^b^ barriers identified from the NSW CATI (see intervention components); ^c^ barriers identified from interventions in ECEC settings [34].

**Table 2 ijerph-20-05043-t002:** Intervention schedule.

Month	Type of Contact	When	Method	Resources
0	Admin	Recruitment	Email	Recruitment pack
0	Admin	Recruitment	Call	Nominated-supervisor recruitment survey
0	Admin	Recruitment	Email	Free-play record
0	Admin	Recruitment	Email	Allocation email Opinion Leader position description
1	Admin	Baseline	Call	Administrative contact to book 1st intervention contact and identify Opinion Leader
1	First intervention contact	One-month timepoint Introductory meeting	Videoconference call	Introductory-meeting script Resource pack Sustainability action plan Guidance for addressing barriers
1	Admin	One-month timepoint after introductory support meeting	Email	Introductory-support-call summary email
1	Second intervention contact	Two-month timepoint Policy Review #1	Email	Service policy Policy template Policy checklist Policy feedback Policy review email
3	Admin	3-month timepoint Reminder email	Email	Three-month timepoint reminder email
3	Third intervention contact	3-month timepoint Support meeting	Videoconference call	Guidance for addressing barriers Sustainability action plan
3	Admin	Three-month timepoint After Support meeting	Email	Three-month support call summary email
4–5	Fourth intervention contact	Four/five-month timepoint Policy Review #2	Email	Service policy Policy template Policy checklist Policy feedback Policy-review email
	Additional contacts	As needed throughout the intervention period	Email/Call/Videoconference call	As needed
6	Follow-up	Six-month post-baseline timepoint End of intervention	Email/Call	Free-play record Sustainability action plan
		As needed throughout the intervention period		As needed
12	Follow up	Twelve-month timepoint	Email/Call	Free-play record Sustainability action plan

**Table 3 ijerph-20-05043-t003:** Feasibility and Acceptability measurement.

Intervention acceptability and feasibility (intervention services only)					
Over the last 6 months, your service was provided with support to continue to provide opportunities for indoor–outdoor free play in an ongoing way, as part of our program known as Sustaining Play, Sustaining Health. We are interested in your experience, and the acceptability and feasibility of the support provided to your service as part of this program. There are no right or wrong answers. We are interested in your beliefs and perceptions. By answering these questions, you will help us to identify what support may assist services in the ongoing delivery of health programs. We will ask you to answer each question using a scale from 1 to 5.					
TFA construct	Generic TFA questionnaire items				
Affective attitude How you felt about the program	A1. Did you like or dislike the Sustaining Play, Sustaining Health program?				
	Strongly dislike	Dislike	No opinion	Like	Strongly like
	1	2	3	4	5
Burden The amount of effort required to participate in the program	A2. How much effort did it take to engage with the Sustaining Play, Sustaining Health program?				
	No effort at all	A little effort	No opinion	A lot of effort	Huge effort
	1	2	3	4	5
Perceived effectiveness The extent to which the prograe is perceived to have achieved its objective	A3. The Sustaining Play, Sustaining Health program has helped to continue the sustainment of indoor–outdoor free-play routines.				
	Strongly disagree	Disagree	No opinion	Agree	Strongly agree
	1	2	3	4	5
Intervention coherence The extent to which the participant understands how the program works	A4. It is clear to me how the Sustaining Play, Sustaining Health program has helped to continue the sustainment of indoor–outdoor free-play routines at my service.				
	Strongly disagree	Disagree	No opinion	Agree	Strongly agree
	1	2	3	4	5
	Please tell us more about your views				
Self-efficacy A participant’s confidence that they can perform task(s) required to participate in the program	A5. How confident did you feel about participating in the Sustaining Play, Sustaining Health program?				
	(i.e., How confident did you feel about using the sustainability strategy to continue to deliver indoor-outdoor free play routines at your service?)				
	Very unconfident	Unconfident	No opinion	Confident	Very confident
	1	2	3	4	5
Opportunity costs The benefits, profits or values that would have to be given up to engage with the program	A6. Participating in the Sustaining Play, Sustaining Health program interfered with my other priorities				
	(i.e., Did implementing the sustainability strategy interfere with your other priorities?)				
	Strongly disagree	Disagree	No opinion	Agree	Strongly agree
	1	2	3	4	5
General acceptability	A7. How acceptable was the Sustaining Play, Sustaining Health program to you?				
	Completely unacceptable	Unacceptable	No opinion	Acceptable	Completely acceptable
	1	2	3	4	5
A8. Is there anything further you would like to mention regarding how acceptable the Sustaining Play, Sustaining Health program was at your service?					
Feasibility	F1. Continuing to implement the sustainability strategy used in the Sustaining Play, Sustaining Health program seems possible.				
	Strongly disagree	Disagree	No opinion	Agree	Strongly agree
	1	2	3	4	5
	F2. Continuing to implement the sustainability strategy used in the Sustaining Play, Sustaining Health program seems doable.				
	Strongly disagree	Disagree	No opinion	Agree	Strongly agree
	1	2	3	4	5
	F3. Continuing to implement the sustainability strategy used in the Sustaining Play, Sustaining Health program seems easy.				
	Strongly disagree	Disagree	No opinion	Agree	Strongly agree
	1	2	3	4	5
F4. Is there anything further you would like to mention regarding how feasible the Sustaining Play, Sustaining Health program was at your service?					

## Data Availability

This study is a study protocol, and no data are generated as part of the manuscript. Data sharing is not applicable to this article as no datasets were generated or analysed during the current study. Trial results will be disseminated via publication in an academic journal, presentation at national and international conferences, as part of PhD student thesis and disseminated to ECEC services and to relevant end-user agencies.

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
