# Peer review of "A Pilot Randomised Controlled Trial to Increase the Sustainment of an Indoor–Outdoor-Free-Play Program in Early Childhood Education and Care Services: A Study Protocol for the Sustaining Play, Sustaining Health (SPSH) Trial"

_ijerph, 2023, doi:10.3390/ijerph20065043_

Round 1
Reviewer 1 Report
Overall, this is a very well-written study protocol, comprehensive. Only a minor comments, in 2.2.3 and 2.4.2, it was mentioned about recruiting 20 ECEC, there is no reference used to support this figure. Please add.
In 2.5.1, I just wondering did authors simply assume data will be normally distributed, thus will be presented parametrically. Why don't author run normality test?
Author Response
Please see the attachment under "Reviewer 1".

Reviewer 2 Report
Dear authors,
Thank you very much for the relevant topic presented in this paper. It describes well the process to develop different strategies to improve health in specific populations. However, I do have some questions or doubts about the paper and believe that some improvements might be addressed in order to contribute to a better quality for the manuscript to be published.
Main concerns
Pag. 15. 2.4.3 Interventions components. Authors describes very well who will deliver the intervention. In addition, it is stated that one of the objectives is assess acceptability, feasibility and impact of the SPSH program on the sustainment of indoor-outdoor free play programs, and these programs are addressed to improve PA levels of promote children to be physically active. Thus, we believe that specialists on physical activity and exercise might be part of the delivering teams. More information should be added in this regard.
Pag. 19. 2.5 Data collection and measures. The persons responsible for reporting the feasibility and acceptability of the SPSH program have been previously trained to do it properly. Did they receive information about how to score the indices of acceptability or feasibility?. More information about this need to be addressed.
Pag. 32. References. Many references are written in the list of references, however, they are not cited in the text of the manuscript.
Minor concerns
Pag 2. Line 55 and 57. “This is can be defined”, this sentence needs to be revised, there is a grammar error.
Pag. 5. Table 1. The design and style of the table is very wide and with long cells. It may be revised to facilitate the view and reading of this table, i.e. reducing font sizes or between line spaces.
Pag. 15. Footnote c. Add a blank space between settings and (32)
Pag. 17. Table 2. Underlined words
Pag 5. Table 1 and Table 2. Different types of fonts are shown in the cells, in addition, the design and style of tables should be similar. Underlined words may be not underlined.

Author Response
Please see the attachment under "Reviewer 2"

Round 2
Reviewer 2 Report
Dear authors,
Thank you very much for the effort to address all recommendations and comments
Minor concerns
Pag. 4. Tables design. The reviewer encouraged authors to revise the structure of tables to facilitate legibility, i.e. reducing font sizes or between line spaces. Although the authors mention that the tables have been amended and added in a separate file, we could not see that attached file to revise it again.
Author Response
Minor concerns
Pag. 4. Tables design. The reviewer encouraged authors to revise the structure of tables to facilitate legibility, i.e. reducing font sizes or between line spaces. Although the authors mention that the tables have been amended and added in a separate file, we could not see that attached file to revise it again.
Authors response: Thank you for informing us that you were unable to view the attached file. We have now uploaded the file in hopes that you would be able to view it and revise it.